# Simulated Microgravity Exposure Induces Antioxidant Barrier Deregulation and Mitochondria Enlargement in TCam-2 Cell Spheroids

**DOI:** 10.3390/cells12162106

**Published:** 2023-08-19

**Authors:** Marika Berardini, Luisa Gesualdi, Caterina Morabito, Francesca Ferranti, Anna Reale, Michele Zampieri, Katsiaryna Karpach, Antonella Tinari, Lucia Bertuccini, Simone Guarnieri, Angela Catizone, Maria A. Mariggiò, Giulia Ricci

**Affiliations:** 1Department of Anatomy, Histology, Forensic-Medicine and Orthopedics, Section of Histology and Embryology, “Sapienza” University of Rome, 00161 Rome, Italy; marika.berardini@uniroma1.it (M.B.); luisa.gesualdi@uniroma1.it (L.G.); angela.catizone@uniroma1.it (A.C.); 2Department of Neuroscience, Imaging and Clinical Sciences-CAST, “G. d’Annunzio” University of Chieti-Pescara, 66013 Chieti, Italy; cmorabit@unich.it (C.M.); simone.guarnieri@unich.it (S.G.); maria.mariggio@unich.it (M.A.M.); 3Human Spaceflight and Scientific Research Unit, Italian Space Agency, 00133 Rome, Italy; francesca.ferranti@asi.it; 4Department of Experimental Medicine, “Sapienza” University of Rome, 00161 Rome, Italy; anna.reale@uniroma1.it (A.R.); michele.zampieri@uniroma1.it (M.Z.); karpach.1792414@studenti.uniroma1.it (K.K.); 5Center for Gender-Specific Medicine, Gender Prevention and Health Section, ISS Istituto Superiore di Sanità, 00161 Rome, Italy; antonella.tinari@iss.it; 6Core Facilities, ISS Istituto Superiore di Sanità, 00161 Rome, Italy; lucia.bertuccini@iss.it; 7Department of Experimental Medicine, Università degli Studi della Campania “Luigi Vanvitelli”, 80138 Naples, Italy

**Keywords:** mitochondria, simulated microgravity, cellular spheroids, TCam-2 cells, oxidative stress, antioxidant barrier

## Abstract

One of the hallmarks of microgravity-induced effects in several cellular models is represented by the alteration of oxidative balance with the consequent accumulation of reactive oxygen species (ROS). It is well known that male germ cells are sensitive to oxidative stress and to changes in gravitational force, even though published data on germ cell models are scarce. We previously studied the effects of simulated microgravity (s-microgravity) on a 2D cultured TCam-2 seminoma-derived cell line, considered the only human cell line available to study in vitro mitotically active human male germ cells. In this study, we used a corresponding TCam-2 3D cell culture model that mimics cell–cell contacts in organ tissue to test the possible effects induced by s-microgravity exposure. TCam-2 cell spheroids were cultured for 24 h under unitary gravity (Ctr) or s-microgravity conditions, the latter obtained using a random positioning machine (RPM). A significant increase in intracellular ROS and mitochondria superoxide anion levels was observed after RPM exposure. In line with these results, a trend of protein and lipid oxidation increase and increased pCAMKII expression levels were observed after RPM exposure. The ultrastructural analysis via transmission electron microscopy revealed that RPM-exposed mitochondria appeared enlarged and, even if seldom, disrupted. Notably, even the expression of the main enzymes involved in the redox homeostasis appears modulated by RPM exposure in a compensatory way, with GPX1, NCF1, and CYBB being downregulated, whereas NOX4 and HMOX1 are upregulated. Interestingly, HMOX1 is involved in the heme catabolism of mitochondria cytochromes, and therefore the positive modulation of this marker can be associated with the observed mitochondria alteration. Altogether, these data demonstrate TCam-2 spheroid sensitivity to acute s-microgravity exposure and indicate the capability of these cells to trigger compensatory mechanisms that allow them to overcome the exposure to altered gravitational force.

## 1. Introduction

Crewed long-lasting space flights, to the Moon and Mars, represent one of the most ambitious challenges in the present and near future, and are predicted to have a huge strategic impact on the development of advanced technologies [1]. However, space biomedicine research since the beginning of the space exploration era [2] has revealed that the space environment is hostile for human beings, and in general for all living systems [3,4]. Recent reviews [2,4] have clearly indicated the five threats to human health related to long-lasting spaceflights, and one of these is microgravity exposure. A full knowledge of the effects of microgravity is necessary to develop the proper countermeasures that can limit the physiology alterations of spaceflight crew. It is fair to highlight that the data on the effects of real spaceflight are limited, and most of our knowledge on the influence of microgravity on living systems has been obtained thanks to ground-based microgravity simulators that allow the total control of the applied gravitational conditions, the simulation of different levels of gravitational force, as well as the relatively low-cost replication of the experiments [5]. To date, several ground-based microgravity simulators have been developed [6], and one of them is the random positioning machine (RPM), which was chosen in the present study to mimic microgravity conditions.

Published data reveal that real or simulated microgravity can affect the physiology of cells, interfering with different cell mechanic “sensors” that differ depending on the phenotypic features and differentiation status of the cells [7]. However, it seems that a common signature of the effect of microgravity on mammalian cells is represented by the alteration of oxidative metabolism, with an increase in ROS levels [8,9]. Among the sources of ROS in eukaryotic cells, the main systems are the mitochondrial electron transport chain and the NADPH oxidases (NOXs). These enzymatic systems control the balance among oxidative eustress and oxidative distress, and a complex crosstalk and substantial interplay between mitochondria and NOX activity have been demonstrated in different cellular models [10]. It is worth mentioning that ROS formation is necessary under physiological conditions, being active as second messengers and signal transduction molecules (oxidative eustress) that regulate cellular processes such as survival, growth, proliferation, and apoptosis [11]. On the other hand, it is well known that the excessive production of ROS results in oxidative stress (or oxidative distress), which in turn causes cellular dysfunctions or damage to cell biomolecules, including proteins, lipids, and nucleic acids.

To maintain the oxidative balance and avoid the harmful effects of ROS accumulation, mammalian cells are provided by sophisticated antioxidant systems. These systems include antioxidant enzymes such as superoxide dismutase (SOD), catalase (CAT), and glutathione peroxidase (GPX1), as well as non-enzymatic antioxidants such as vitamin C and E and glutathione [12]. The mitochondria themselves, considered the main endogenous producers of ROS, possess their own antioxidant defense systems. Furtherer mitochondria have the ability to modulate cellular H_2_O_2_ levels, thus functioning as key regulators of ROS-mediated signaling pathways [13].

The balance between ROS generation and antioxidant-mediated ROS detoxification ensures that ROS levels are tightly controlled and finely tuned to act as second messengers and cell signaling molecules: the lack of this coordination results in ROS accumulation and cell damage.

Long-lasting spaceflights make it necessary to answer to the open question regarding the effect of the space environment on male germ cells [14,15], since their deregulation can be inherited by an astronaut’s offspring [16]. Therefore, the reproductive health of spaceflight crews is becoming a crucial issue to addresss, and is still underinvestigated. Published data on the effects of microgravity on human male germ cells are scarce and mainly limited to male gametes (spermatozoa), with variable results [17,18,19]. Animal models indicate that mitotic and post-mitotic germ cells are sensitive to gravitational force [20]. In particular, in mouse spermatogonia in vitro cell cultures, microgravity exposure stimulates differentiation and meiotic entry [20,21]. Moreover, some authors have suggested the impact of simulated microgravity on mouse sperm mitochondria metabolism [22]. It is fair to highlight that in vivo studies and ex vivo organ cultures in microgravity conditions provided evidence of microgravity-triggered alterations in germ cell behavior, even though these studies did not necessarily report the direct effect of microgravity on germ cells, because the alterations of germ cells can be ascribed to the well-known detrimental effect of microgravity on steroidogenic Leydig cells [23,24,25]. Finally, it should be mentioned that germ cells are also known to be very sensitive to oxidative stress, and therefore it is of particular interest to evaluate the effect of microgravity on these cell types, monitoring the key points of their oxidative system [26].

In the present manuscript, we studied the possible microgravity-triggered alterations in a human model of male primordial germ cells: TCam-2 cells [27]. Therefore, the use of this cell line is strategic for the investigation of microgravity’s impact on the stem cells responsible for the self-renewal of male germ cells. In previous papers, we studied the effects of microgravity on these cells cultured as a monolayer (2D cell culture) [28,29], observing microgravity-dependent transient activation of autophagy, intracellular calcium increase, and ROS accumulation. To evaluate the role of adhesive behavior in the cell response to s-microgravity, in this manuscript, we cultured TCam-2 cells in low-adhesion conditions: in this way, the cells do not adhere to a substrate but are forced to maintain contact with each other, forming cell-to-cell junctions and giving rise to TCam-2 cell 3D spheroids. These samples were exposed to RPM for 24 h, which is the culture time in which we found major responses to s-microgravity in the 2D TCam-2 cell cultures.

## 2. Materials and Methods

### 2.1. Cell Culture and Microgravity Parameters

TCam-2 seminoma cells were cultured in RPMI 1640 (Sigma-Aldrich, cat. R8758, St. Louis, MO, USA) supplemented with 10% fetal bovine serum (FBS, Gibco, cat. 10270, Gland Island, NY, USA), penicillin/streptomycin (Sigma-Aldrich, cat. P0781, St. Louis, MO, USA), L-glutamine (Sigma Aldrich, cat. G7513, St. Louis, MO, USA), gentamicin (Sigma Aldrich, cat. G1272, St. Louis, MO, USA) and MEM NEAA (Gibco, cat. 11140-035, Gland Island, NY, USA) (see Ferranti et al. 2014 for details [29]). Mycoplasma testing was routinely performed with the N-GARDE Mycoplasma PCR Reagent set (Euro-Clone, cat. EMK090020, Milano, Italy).

To obtain TCam-2 spheroids, 60,000 cells/cm^2^ were seeded in low-attachment Petri dishes with coating agar 1% in RPMI and 10% FBS. TCam-2 cells were seeded for 24 h at 37 °C in a humidified atmosphere with 5% CO_2_. When the spheroids were completely formed, medium was added up to the lid to avoid air bubble formation. The medium used for TCam-2 spheroid cultures was the same as that which we used for TCam-2 2D cell cultures.

Then, experiments were performed on spheroids cultured for 24 h in the presence of s-microgravity in the RPM or at 1 g (Ctr) in the same incubator. This incubation time was chosen to test the possible cellular responses to acute exposure, also considering that the 2D TCam-2 cultures regained their main features after a 48 h exposure time under s-microgravity conditions [28].

The RPM was a 3D clinostat. It consisted of two rotating frames. This tool does not eliminate gravity, but is based on the principle of “gravity vector averaging”: it allows one to apply a 1 g stimulus in all directions and the sum of gravitational force vectors reaches zero. All experiments were planned as described in [28].

### 2.2. Cell Viability Assay

To reveal spheroids’ viability, we performed adhesion tests on plastic.

After 24 h of unitary gravity or RPM exposure, TCam-2 spheroids were placed at 1 g in a Petri dish and cultured for 24 h at 37 °C in a humidified atmosphere with 5% CO_2_. Then, in order to analyze spheroid adhesion, the samples were observed with an optical microscope and images were recovered using a 10× obj. (Leica microscope DM IRB, Mannheim, Germany). To confirm spheroids’ viability, we used an Apoptosis/Necrosis assay kit (Abcam, cat. ab176749, Cambridge, UK). Live TCam-2 spheroids after 24 h of Ctr and RPM were stained with Apopxin green indicator (apoptotic cells), 7-AAD (necrotic cells), and CytoCalcein violet 450 (healthy cells) according to the manufacturer’s instructions and incubated at room temperature for 45 min. After staining, spheroids were analyzed using a confocal microscope (Zeiss LM900 confocal microscope, Zeiss, Oberkochen, Germany). Optical spatial series with a step size of 1 µm were recovered. The intensity of CytoCalcein violet, Apopxin green, and 7-AAD red fluorescences was determined with the use of Zen 3.0 Blue Edition software(Jena, Germany), using the sum of intensity (Sum(I)), which represents the total fluorescent intensity recovered within the z-axis of each series. The quantitative data for each spheroid stack profile were normalized for total area. The means ± S.E.M. for green (apoptotic cells) and red fluorescence (necrotic cells) were graphed as the ratio between Sum(I)/µm^2^ values.

### 2.3. Measurements of Glucose and Lactate Levels

The measurements of glucose and lactate levels in the growth media were performed according to the protocol described in Morabito et al. [28]. To normalize the results, glucose levels were expressed as g of glucose in the medium and mg of proteins from the spheroids; lactate levels were expressed as mmol of lactate in the medium and mg of proteins from the spheroids.

### 2.4. Western Blot Analysis

Proteins were extracted from TCam-2 spheroids recovered after 24 h of Ctr or RPM, quantified and separated using a protocol previously described by Morabito et al. [28]. Equal amounts of proteins (20 µg) were loaded for Western blotting analysis. The membranes were blocked with EveryBlot Blocking Buffer (Biorad, Segrate, Italy) for 15 min at room temperature before incubation with primary antibodies overnight at 4 °C. After washing, the membranes were incubated with horseradish peroxidase-conjugated appropriate secondary antibodies (1:10,000 in blocking buffer) for 1 h at room temperature. The signals were detected using an ECL kit (GE Healthcare, Cologno Monzese, Italy) and analyzed with an image acquisition system (Uvitec mod Alliance 9.7, Uvitec, Cambridge, UK). Primary antibodies used in this study included: a mouse monoclonal anti calmodulin (Merck Life Science S.r.l., Milan, Italy, cod 05-173 1:1000 dilution), a rabbit monoclonal anti- phospho-CaMKII (Thr286) (Cell Signaling Technology, Pero, Italy, cod. 12716 1:1000 dilution), a rabbit monoclonal anti-CaMKII (Cell Signaling Technology, cod. 4436 1:1000 dilution), a rabbit polyclonal anti ATP2A2/SERCA2 (Cell Signaling Technology, cod. 4388 1:1000 dilution), a rabbit polyclonal anti-TOMM 20 (Thermo Fisher Scientific, Monza, Italy, cod. PA5-52843, 1:1000 dilution); a rabbit polyclonal anti-SOD1 (Thermo Fisher Scientific, cod. PA527240, 1:5000 dilution,); a mouse monoclonal anti-SOD2 (Thermo Fisher Scientific, cod. MA5-31514, 1:5000 dilution); a rabbit monoclonal anti-catalase (Cell Signaling Technology, Pero, Italy, cod. 14097, 1:1000 dilution); a rabbit polyclonal anti-GPX1 (Thermo Fisher Scientific, cod. PA526323, 1:1000 dilution); a mouse monoclonal anti-NOX2 (Santa Cruz Biotechnology Inc., SantaCruz, CA, USA, cod. sc-130543, 1:500 dilution); rabbit monoclonal antibody anti- NOX4 (Thermo Fisher Scientific, cod. MA5-32090, 1:1000 dilution); a mouse monoclonal anti-4HNE (Merck Life Science S.r.l., Milan, Italy, cod. SAB5202472, 1:1000 dilution); a polyclonal rabbit anti-3-nitrotyrosine (Merck Life Science S.r.l., cod. 4511, 1:1000 dilution); and a polyclonal rabbit anti-LC3B (Merck Life Science S.r.l., cod. L7543, 1:1000 dilution). A mouse monoclonal anti-GAPDH antibody (Merck Life Science S.r.l, 1:10,000 dilution) was used as a loading control. The choice of GAPDH as the loading control was driven by published data revealing a stable expression of this enzyme under microgravity conditions in different cell models [28,30,31,32,33,34,35,36].

### 2.5. Transmission Electron Microscopy Analysis

To evaluate cells’ fine sub-cellular ultrastructural features, samples were fixed in glutaraldehyde 2.5% and embedded in resin for transmission electron microscopy purposes. Briefly, fixed samples were rinsed with cacodylate buffer for at least 1 h, post-fixed with 1% OsO_4_ in a cacodylate buffer, dehydrated in ethanol, and embedded in epoxy resin. Ultrathin sections (60 nm) were treated with uranyl-acetate and then contrasted with lead hydroxide. Samples were studied using a transmission electron microscope, EM208S PHILIPS (FEI—Thermo Fisher), and the mitochondria width (that is the mitochondria minor axis) of the Ctr and RPM-exposed samples was measured.

### 2.6. Confocal Live Cell Imaging Analysis for Mitochondria Function

#### 2.6.1. JC1 Assay

To reveal mitochondrial membrane potential, we used a kit based on the JC1 indicator (Invitrogen, cat. T3168, Carlsbad, CA, USA) on live TCam-2 spheroids after 24 h of Ctr and RPM according to the manufacturer’s instructions. JC1 dye exhibits potential-dependent distribution on the sides of the mitochondrial membrane, indicated by a green fluorescence emission (at 530 nm) for the monomeric form, which shifts to a red one (at 590 nm) with a concentration-dependent formation of J-aggregates. Indeed, mitochondria depolarization is indicated by a decrease in the red/green fluorescence ratio.

The analyses were performed using a confocal microscope (Zeiss LM900 confocal microscope). Optical spatial series with a step size of 1 µm were recovered. The intensity of the red and green fluorescence was determined with the use of Zen 3.0 Blue Edition software, using the sum of intensity (Sum(I)). The quantitative data for each spheroid stack profile were normalized for total area and means ± S.E.M. was graphed as ratio between red fluorescence and green fluorescence.

#### 2.6.2. Mito SOX Assay

To reveal mitochondrial superoxide anions, we used MitoSOX^TM^ Red reagent (Invitrogen, cat. M36008, Carlsbad, CA, USA) on live TCam.2 spheroids after 24 h of Ctr and RPM. Cell staining was prepared according to the manufacturer’s instructions. This reagent is specifically oxidized by mitochondrial superoxide anions, and not by other ROS or RNS, becoming highly fluorescent (Em 580 nm). The analyses were performed using a confocal microscope (Zeiss LM900 confocal microscope). Optical spatial series with a step size of 1 µm were recovered. The intensity of the red fluorescence was determined with the use of Zen 3.0 Blue Edition software, using the sum of intensity of red fluorescence (Sum(I)). The quantitative data for each spheroid stack profile were normalized for total area and means ± S.E.M. were graphed as the ratio between Sum(I)/µm^2^ values.

#### 2.6.3. Reactive Oxygen Species Detection

To detect global levels of ROS, we used a detection kit on live cells (Immunological Sciences, cat. ROS-100, Rome, RM, Italy). TCam-2 spheroids were recovered after 24 h of Ctr and RPM. Spheroids were washed using Hank’s balanced salt solution. H2DCFDA was used as an ROS marker and a “Reactive Oxygen Control” was used as a positive inducer of ROS according to the manufacturer’s instructions. After staining, fluorescent products (Em 530) relative to intracellular ROS level were detected using a confocal microscope (Zeiss LM900). Optical spatial series with a step size of 1 µm were recovered. The intensity of the green fluorescence was determined with the use of Zen 3.0 Blue Edition software, using the sum of intensity of green fluorescence (Sum(I)). The quantitative data for each spheroid stack profile were normalized for total area and the means ± S.E.M. were graphed as the ratio between Sum(I)/µm^2^ values.

### 2.7. RNA Isolation, RT-PCR and qRT-PCR Analyses

To evaluate gene expression analysis of the antioxidant barrier enzymes, frozen TCam-2 spheroids (approximately 10^6^ cells) were thawed on ice and processed for RNA extraction by using the RNeasy Mini Kit (Qiagen, Hilden, Germany) according to the instructions of the manufacturer. DNA contamination was eliminated via RNase-free DNase (Qiagen) treatment. The concentration, purity, and integrity of RNA samples were assessed via UV absorbance measurements with a NanoDrop spectrophotometer (Thermo Scientific, Waltham, MA, USA) and agarose gel electrophoresis. Reverse transcription of 2 µg RNA was carried out with the FastGene Scriptase Basic cDNA Synthesis kit (Nippon Genetics Europe, Düren, Germany) using random and oligo(dT) primers.

The mRNA levels were determined by means of quantitative PCR performed by using the qPCRBIO SybGreen mix (PCR Biosystems, London, UK). Reactions were carried out on cDNA corresponding to 30 ng of total RNA in the presence of transcript-specific primers (4 µM) in a final volume of 20 µL. The thermal protocol consisted of a 2 min step at 95 °C, 40 cycles of 5 s at 95 °C and 30 s at 60 °C, followed by melting curve analysis. Amplifications were run in triplicate on the iCycler IQ detection system (BIO-RAD, Hercules, CA, USA).

Gene expression was quantified by means of the comparative CT method using the expression level of the β-glucuronidase gene (GUSB) as a reference. An inter-run calibration sample was used in each plate to correct for technical variance between runs and to compare results from different plates. The calibrator consisted of cDNA prepared from HEK293T cells. In each experimental PCR plate, samples were measured in triplicate. The primers used in the assay are described in Appendix A.

### 2.8. Statistical Analyses

Statistical analysis was performed using the GraphPad Prism 8.0.2 software for Windows (GraphPad Software Inc., San Diego, CA, USA). All data were expressed as means ± S.E.M. Statistical comparisons between groups were performed using Student’s *t*-tests or one-way ANOVA tests followed by Bonferroni post hoc tests. The results’ analyses were performed for different independent experiments (n). A *p* value of <0.05 was considered statistically significant. (* *p* < 0.05; ** *p* < 0.01; *** *p* < 0.001). A *p* value of >0.05 was considered not statistically significant (*p* = n.s.).

## 3. Results

After obtaining 3D spheroids from TCam-2 cells, as described in the Materials and Methods section, different physiological parameters were evaluated, for the purposes of investigating the cellular response to s-microgravity conditions.

### 3.1. TCam-2 Spheroids’ Viability Is Preserved during Acute Exposure to Simulated Microgravity

The viability of TCam-2 spheroids cultured for 24 h at unitary gravity (Ctr) or in simulated microgravity (s-microgravity) under RPM exposure was assessed by means of the adhesion test on plastic. All spheroids had the ability to adhere on plastic and to form viable cell clusters, irrespective of culture condition. Representative images obtained after this qualitative analysis are shown in Figure 1A.

To further characterize cell viability, we used the Apoptosis/Necrosis assay kit that allowed us to evaluate the presence of necrotic and apoptotic cells. We performed fluorescence staining for 7-amino-actinomycin-D (7-ADD), a marker used to highlight necrotic cells, and APOPXIN, a marker of apoptotic cells. The quantitative analysis performed using confocal microscopy did not show significant differences in terms of the necrotic and apoptotic index in both experimental conditions, and the fluorescence of CytoCalcein-treated cells confirms that TCam-2 spheroids are healthy and alive (Figure 1B).

### 3.2. Metabolic Features (Glucose Consumption and Lactate Production) Are Not Significantly Altered by Simulated Microgravity

Glucose and lactate levels were measured in the conditioned medium of TCam-2 spheroids after 24 h of culture in Ctr or RPM conditions. These parameters can provide insights into the metabolic pathways of this cellular model. The data reported in Figure 2 show that simulated microgravity does not significantly affect glucose consumption and lactate production in TCam-2 spheroids.

### 3.3. Intracellular Calcium Handling Is a Landmarks in the Pathways Activated by Simulated Microgravity

The 2D TCam-2 cultures exposed to simulated microgravity showed increased intracellular calcium levels after 24 h. To test the involvement of this intracellular signal in the responses of 3D spheroids to these conditions, the expression levels of three of the main keystone proteins (calmodulin, pCaMKII and SERCA) in intracellular calcium handling were assayed after 24 h of exposure. No differences in the expression levels of calmodulin and SERCA2 were observed between exposed (RPM) and control (Ctr) spheroids, as revealed by Western blot analyses. Notably, increased phosphorylated CAMKII levels were observed in exposed samples in comparison to control ones (Figure 3).

### 3.4. Mitochondria’s Ultrastructure and Functional Features Appeared to Be Modified after 24 h of Exposure to Simulated Microgravity

The ultrastructure of TCam-2 spheroids was investigated by using transmission electronic microscopy (TEM). From this ultrastructural analysis, it emerged that in samples subjected to s-microgravity for 24 h (RPM), mitochondria were altered in morphology, showing an increased size (mitochondria width mean values: Ctr = 0.26 μm ± 0.012 S.E.M.; RPM = 0.56 μm ± 0.031 SE.M.; Student’s *t*-test *p* < 0.001), and, occasionally, the membranes of mitochondria were damaged. Notably, in spite of the increased size, mitochondria cristae appeared in an orthodox state, and there was no significant change in the electron density of the mitochondria matrix (Figure 4A). Interestingly, the presence of swollen mitochondria does not influence total mitochondria mass since the mitochondria marker TOMM20 does not appear to be affected by microgravity (Figure 4B).

To test some of the functional features of mitochondria, their membrane potential and ROS generation were analyzed. We used JC-1 dye, an indicator of mitochondrial membrane potential and, as consequence, of mitochondrial activity, in living cells (Figure 5A). Changes in mitochondrial potential, associated with the opening of mitochondrial permeability transition pores and with a decoupling of the respiratory chain, can be monitored using a dual form of JC1 dye. In fact, at a higher membrane potential, which indicates active mitochondria, JC1 forms red fluorescent “aggregates” within the mitochondria. A lower mitochondria membrane potential, which indicates inactive mitochondria, is characterized by dye monomers characterized by green fluorescence. The quantitative analysis of red and green fluorescence was determined via confocal microscopy. By measuring the ratio between red and green fluorescence, it is possible to evaluate the mitochondrial activity of cell spheroids. In fact, mitochondrial depolarization is indicated by a decrease in the red/green fluorescence intensity ratio, whereas an increase is indicative of hyperpolarization. The analyses carried out on TCam-2 spheroids after 24 h of culture under Ctr or RPM conditions did not show significant differences in terms of mitochondrial activity (Figure 5B).

To evaluate the mitochondria’s superoxide anion production, we used MitoSOX™ fluorescence dye. The oxidation of MitoSOX™ in living cells by superoxide anions produces red fluorescence. Simulated microgravity induces superoxide anion production in TCam-2 spheroids, and confocal analysis shows a significant increase in red fluorescence in RPM samples compared with samples cultured at unitary gravity (Figure 6).

### 3.5. Simulated Microgravity Affects Oxidative Balance in TCam-2 Spheroids

The TCam-2 spheroids exposed to s-microgravity showed a significant increase in the cellular ROS levels (Figure 7A). In order to elucidate the oxidative metabolism of TCam-2 spheroids, the expression of the ROS-generating system and the cell antioxidant machinery was investigated using qRT-PCR. Xanthine dehydrogenase (XDH) and the components of the NADPH oxidases complex, CYBA, CYBB, NCF1 and NCF2, representing genes implicated in ROS production in cells were analyzed. We observed reduced expression levels of the catalytic subunit of NOX2 (CYBB) and NCF1 mRNAs (Figure 7D, *p* = 0.012, *p* = 0.027, respectively), indicating a response of the cells in counteracting the increase in ROS concentration in the cells. Notably, no significant changes were observed in the NOX2 protein expression level (Figure 7C).

Interestingly, Western blot analysis revealed that NOX4 is increased under s-microgravity conditions (Figure 7C), indicating that this enzyme can be at least in part responsible of the ROS increase under microgravity conditions.

The analysis of antioxidant genes mRNA level included SOD1, CAT, GPX1, and HMOX1, all involved in cellular protection from oxidative stress. GPX1 enzyme appeared to be downregulated and upregulated after RPM exposure, whereas HMOX1 enzyme appeared to be upregulated. SOD1 and catalase appeared to be unaffected (Figure 8A). In line with these results, even Western blot analysis revealed that GPX1, which detoxifies cells from hydrogen peroxide, is downmodulated after 24 h of microgravity exposure, whereas SOD1 and SOD2, which convert cytoplasmic and mitochondrial superoxide anions into hydrogen peroxide, and catalase, which, as well as GPX1, neutralizes hydrogen peroxide, did not show significant changes (Figure 8B–D). The increase in NOX4 and the decrease in GPX1 expression suggest that exposure to s-microgravity could impair the ability of cells to neutralize ROS, leading to increased oxidative stress and cellular damage over time. Notably, the expression of HMOX, involved in the heme catabolism of mitochondria cytochromes, is increased, as well as the expression of the transcription factor responsible for HMOX1 transcription (NRF2). As HMOX1 is responsible for the production of antioxidant catabolites, its upregulation can be considered a sign of the rescue attempt of redox balance as a consequence of microgravity-induced ROS increase.

### 3.6. Oxidative Damage and Autophagy Are Not Influenced by Simulated Microgravity Exposure

To assess whether exposure to s-microgravity could induce damage to proteins and lipids, the expression of 3-nitrotyrosine and 4-hydroxynonenal, which are markers of protein and lipid oxidation, respectively, was measured. In the samples exposed to simulated microgravity, there was no significant increase in these markers of oxidative stress, although their slight increase indicates an oxidative imbalance (Figure 9A,B). Indeed, ultrastructural observations show the presence of numerous autophagic vesicles in both the control and simulated-microgravity-exposed samples (Figure 9C). However, alongside the structural changes to the cytoskeleton and mitochondria, the autophagic process implies a very complex molecular network in which LC3β plays a crucial role [37]. For this reason, we tested the expression levels of LC3β in our model, also considering that in our previous studies, LC3β was found to be upregulated in 2D TCam-2 cells exposed to s-microgravity [28,29]. In addition, LC3β was also upregulated in other cell models exposed to simulated or real microgravity, indicating that it can represent a good marker to evaluate a possible induction of the autophagic process during altered gravity challenges [34,38]. In line with the results coming from ultrastructural analyses, LC3β protein expression also does not show significant difference between 3D TCam-2 cultured control and RPM-exposed cells (Figure 9D).

## 4. Discussion

Long-lasting crewed space missions are considered one of the major challenges in the near future for human beings. However, our knowledge on the impact of the space environment on human health still has gaps to be filled for the setting up of proper countermeasures counteracting space-related threats. Among them, microgravity represents one of the well-known threats for human health, associated with human space exploration [2], and the protection of human reproductive health in the space environment is fundamental when considering astronaut’s offspring and, possibly, the colonization of Mars. Mitotically active male germ cells have been demonstrated to be responsive to changes in gravitational force [14,15,20]. Previous studies by our group demonstrated the microgravity-triggered responses of TCam-2 cells, a seminoma-derived cell line that maintains several features of primordial germ cells. In those studies, cells were seeded on plastic or glass dishes as monolayers [28,29], and therefore we wondered whether an in vitro culture in three-dimensional conditions, mimicking cell–cell contacts in organ tissue, could modify cell responses to s-microgravity.

The experimental evidence on 2D TCam-2 cells revealed that 24 h of exposure to simulated microgravity induced a slowdown of the cell cycle accompanied by altered oxidative metabolism and intracellular calcium handling. All these effects were transient because they were not evident after 48 h of exposure to simulated microgravity [28]. These findings determined the choice of the experimental plan and timing for the 3D cultures in the present study.

Herein, we report that TCam-2 cell spheroids respond to s-microgravity in the same way as their bi-dimensionally cultured counterparts in some features, even though significant differences in cell response can be ascribed to the different culture conditions such as cell-to-cell and cell–substrate adhesion. Cell death analysis revealed that RPM exposure for 24 h does not influence cell survival, irrespective of the bi- or three-dimensional culture condition, confirming the capability of TCam-2 to resist acute exposure to microgravity (Figure 1) [29]. Moreover, s-microgravity-exposed TCam-2 cell spheroids increase ROS levels and mitochondria superoxide anion production apparently without perturbing mitochondria membrane potential as well as the TCam-2 samples cultured as monolayers (Figure 5, Figure 6 and Figure 7) [28]. However, it should be highlighted that JC1 analysis in the 3D culture system revealed high variability in the results that was not observed in bidimensional cultures. The mean values of the JC1 ratio analyzed in TCam-2 cell spheroids tend to increase in s-microgravity conditions, but, due to the high variability, the values are not statistically significant, indicating that 3D cultures are more heterogenous than bidimensional ones (Figure 5).

In both bi-dimensional and three-dimensional cultures, mitochondria appeared swollen and increased in size. However, the mitochondria of TCam-2 cells cultured as monolayers showed electron-clear swollen areas in their matrix after 24 h of exposure to s-microgravity [28], whereas in three-dimensional cultures, the mitochondrial matrix appeared homogeneously electron-dense (Figure 4). These results indicate an apparently milder impact of s-microgravity on mitochondria in cells cultured as three-dimensional structures. According to this hypothesis, the amount of glucose and lactate in the medium does not significantly change in TCam-2 spheroids after 24 h of s-microgravity exposure (Figure 2). Conversely, they appeared at higher levels in the medium of TCam-2 cells exposed to s-microgravity and cultured as a monolayer [28], indicating an s-microgravity-induced increase in anaerobic metabolism in TCam-2 monolayers after RPM exposure that, therefore, does not occur in 3D-cultured samples. In line with what was observed in 2D cultures [28], even if mitochondria appeared enlarged, no significant differences in the amount of the mitochondria marker TOMM20 were observed in the s-microgravity-exposed samples, indicating a non-significant change in the mitochondrial mass (Figure 4) [28]. The microgravity-induced mitochondrial swelling is an interesting matter that deserves further investigations. We can speculate that this mitochondrial modification can be ascribed to the mitochondria’s oxidation observed via the MitoSOX assay, but it may be also caused by the alteration of mitochondrial fission and fusion dynamics, which are also known to be modulated by ROS levels [39].

Trying to explain the source and the consequence of the observed ROS increase, we analyze using qRT-PCR and Western blot analyses the expression of key components of the redox cellular systems. We observed a partial impairment of the antioxidant barrier, since the glutathione peroxidase (GPX1) gene and protein [40] appeared to be downregulated, while SOD 1–2 and catalase were not affected by s-microgravity (Figure 8). Moreover, in line with the ROS increase, we observed a significant increase in NADPH oxidase 4 (NOX4) in TCam-2 spheroids exposed to s-microgravity (Figure 7). Altogether, these results can explain, at least in part, the s-microgravity-triggered ROS increase. Notably, the s-microgravity-induced pro-oxidant environment does not seem passively tolerated by the TCam-2 cell spheroids, as we observed signs of recovery in order to attempt the rescue of redox balance. Indeed, CYBB mRNA, which codifies for the catalytic domain of NOX2, and NCF1 mRNA, which codifies for the cytosolic subunit of neutrophil NADPH oxidase, appeared slightly but significantly reduced in spheroids exposed to s-microgravity (Figure 7). It is fair to highlight that in spite of the decrease in gene transcription, the NOX2 protein level does not seem to be affected by s-microgravity (Figure 7), indicating that the rescue of oxidative balance was not already reached by TCam-2 spheroids. Moreover, we observed the s-microgravity-induced upregulation of the heme oxygenase 1 (HMOX1) gene mRNA, probably as a consequence in turn of NRF2 transcription factor upregulation (Figure 8) [41]. HMOX1 codifies for an enzyme, inducible by oxidative stress, that catalyzes the degradation of heme to form carbon monoxide (CO), iron ions (Fe), and bilirubin. These molecules are considered to have direct or indirect antioxidant properties [42,43], and therefore, in TCam-2 spheroids, the upregulation of HMOX1 can counteract the s-microgravity-triggered oxidation. Moreover, as this enzyme is involved in mitochondrial cytochrome metabolism, it can be also involved in the turnover of the oxidized s-mitochondria [44] that we observed using a MitoSOX assay and ultrastructural analysis as a consequence of s-microgravity exposure (Figure 4 and Figure 6). Notably, NRF2 is a key regulator of transcription of several genes involved in the cell response to oxidative stress, through the binding to a common DNA sequence called antioxidant response element (ARE) [45,46]. Therefore, the upregulation of NRF2 represents a good indicator of the homeostatic broad cellular response to microgravity-dependent oxidative stress.

Notably, even in bidimensional cultures, we observed a protective reaction against ROS increase, since after 24 h of RPM exposure, SOD1 appeared to be increased [28], but, as already mentioned, the levels of this protein are not affected by s-microgravity in 3D spheroids.

In the 2D cultures, we also observed an s-microgravity-triggered increase in intracellular calcium, which is mirrored in the 3D culture system by the increase in the pCAMKII (Figure 3). Interestingly, CAMKII can be activated both by the increase in intracytoplasmic calcium ions, or by the ROS increase [47,48], and therefore represents another sign of the cell’s reaction to ROS induced by RPM exposure. The efficacy of a cell’s counteraction to an ROS increase in TCam-2 cell spheroids is demonstrated by the absence of a significant increase in protein and lipid oxidation of RPM-exposed samples with respect to those cultured at unitary gravity (Figure 9). It should be highlighted that the autophagic process can exert a protective role against oxidation, removing the damaged cellular organelles. Indeed, autophagy induction has been observed after s-microgravity exposure in 2D TCam-2 cell cultures [29]. However, the study of the autophagy marker LC3β via Western blot analysis fails to observe a modulation of autophagy after 24 h of RPM exposure in TCam-2 3D cultures (Figure 9). Interestingly, ultrastructural studies revealed that the autophagy process is already active in TCam-2 spheroids even when cultured at unitary gravity (Figure 9), and therefore, presumably, the ability to rescue against stressors is more pronounced in the TCam-2 three-dimensional cultures than in bidimensional ones.

Nevertheless, it should be considered that autophagy is a complex process induced by a very large number of physiological (as during development) or pathological conditions or stressors. Moreover, it should be highlighted that our results do not exclude the possibility that the active autophagic process could be used by TCam-2 spheroids to rapidly rescue the s-microgravity-induced ROS-dependent cell damages, or that autophagy overactivation occurs at different times during s-microgravity exposure. This scenario paves the way for further investigation in our model in order to distinguish the autophagic features present in the 3D cultures themselves and those induced by s-microgravity exposure.

## 5. Conclusions

Taken together, our results demonstrate that TCam-2 cells are sensitive to short-lasting changes in gravitational forces even when cultured as three-dimensional cell suspensions. Intriguingly, some of the responses to s-microgravity seem to be influenced by the 2D or 3D culture conditions, even if there are several aspects of microgravity-triggered cell responses that are common in both culture conditions. Indeed, 3D TCam-2 spheroids show markers of oxidative stress and altered intracellular calcium handling, all features common to their 2D counterpart in the response to s-microgravity. Moreover, our results highlight the ability of TCam-2 cells to trigger compensatory mechanisms to counteract microgravity-induced cell alterations, and interestingly, the culture condition influences the type of molecular and cellular mechanisms that counteract the s-microgravity triggered stressors, as occurs for the autophagic process.

Nevertheless, the TCam-2 3D model, mimicking an organ-like behavior of a mitotically active germ cell niche, reveals important findings useful for understanding the mechanisms of the effects induced by microgravity on the male reproductive system. In this way, protection strategies can be developed to defend the health of astronauts who experience the space environment, of which microgravity is one of the various stressors. Therefore, these findings could be significant not only for astronauts, but also for their offspring and for future generations, due to microgravity’s impact on gametogenesis.

## Figures and Tables

**Figure 1 cells-12-02106-f001:**
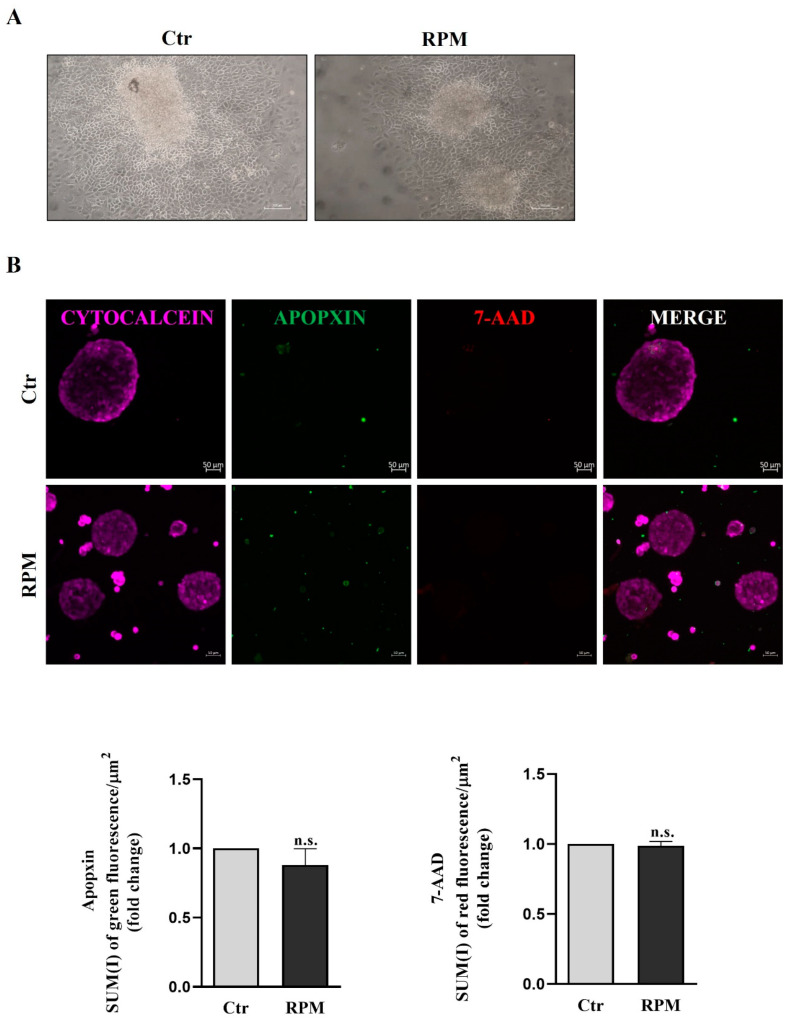
**Microgravity does not significantly affect the cell survival of TCam-2 spheroids.** (**A**) Representative images of the adhesion test of TCam-2 spheroids after 24 h of culture in Ctr or RPM condition is shown. (Scale bar = 100 µm, n = 3 independent experiments). (**B**, **upper panel**) Representative images of TCam-2 spheroids recovered after 24 h of culture in Ctr and RPM conditions stained with CytoCalcein (health cells), 7-ADD (necrotic cells) and Apopxin (apoptotic cells) and analyzed via confocal microscopy (scale bar = 50 µm, n = 3 independent experiments). (**B**, **lower panel**) Quantitative analysis of green fluorescence/area (Apopxin) to evaluate apoptotic cells, and quantitative analysis of red fluorescence/area (7-ADD) to evaluate necrotic cells (means ± S.E.M.; Student’s *t*-test, *p* = n.s.; n = 3 independent experiments).

**Figure 2 cells-12-02106-f002:**
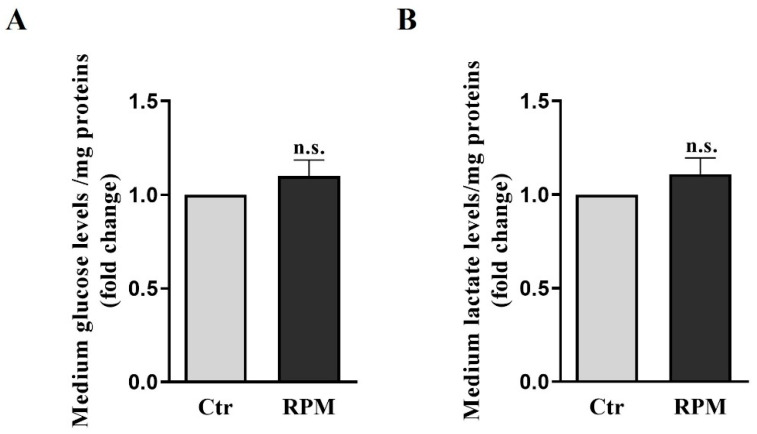
**Glucose and lactate levels in growth medium.** Data are expressed as the ratio between results from samples exposed to s-microgravity (RPM) and those from the corresponding control (Ctr). (**A**) Glucose levels (g glucose/mg proteins). (**B**) Lactate levels (mmol lactate/mg protein) (means ± S.E.M.; Student’s *t*-test, *p* = n.s.; n = 15 independent experiment).

**Figure 3 cells-12-02106-f003:**
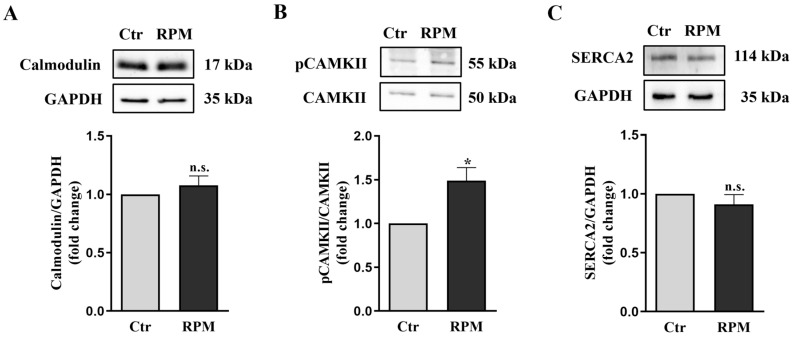
**Expression of calcium handling proteins**: (**A**–**C**) Representative immunoblots of calmodulin, pCaMKII and SERCA2 and the corresponding densitometric analyses. The densitometric values were calculated as the ratio between the optical density (OD) × mm^2^ of each band and the OD × mm^2^ of the corresponding loading control (GAPDH band for Calmodulin and SERCA2; CaMKII for pCaMKII). The data are expressed as the ratio between results from samples exposed to s-microgravity (RPM) and those from the corresponding control (Ctr) (means ± S.E.M.; Student’s *t*-test, in A and C *p*=n.s., in B * *p* < 0.05, n = 6 independent experiments).

**Figure 4 cells-12-02106-f004:**
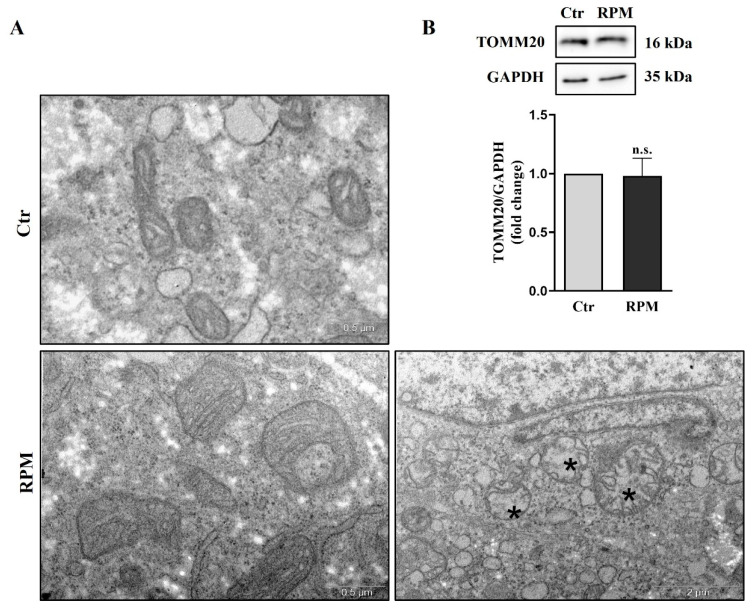
Simulated microgravity significantly affects TCam-2 spheroid mitochondria’s ultrastructure without altering the global mitochondria network mass. (**A**) Representative images of ultrastructural analysis performed using TEM showing mitochondria morphology in Ctr and RPM culture conditions (scale bar of the images on the left of the panel = 0.5 µm; scale bar of the image on the right of the panel = 2 µm. The asterisks indicate damaged mitochondria, n = 3 independent experiments). (**B**) Representative immunoblots of TOMM20 and the corresponding densitometric analysis. The densitometric values were calculated as the ratio between the optical density (OD) × mm^2^ of each band and the OD × mm^2^ of the corresponding loading control (GAPDH band). The data are expressed as the ratio between results from samples exposed to s-microgravity (RPM) and those from the corresponding control (Ctr) (means ± S.E.M.; Student’s *t*-test, *p* = n.s., n = 6 independent experiments).

**Figure 5 cells-12-02106-f005:**
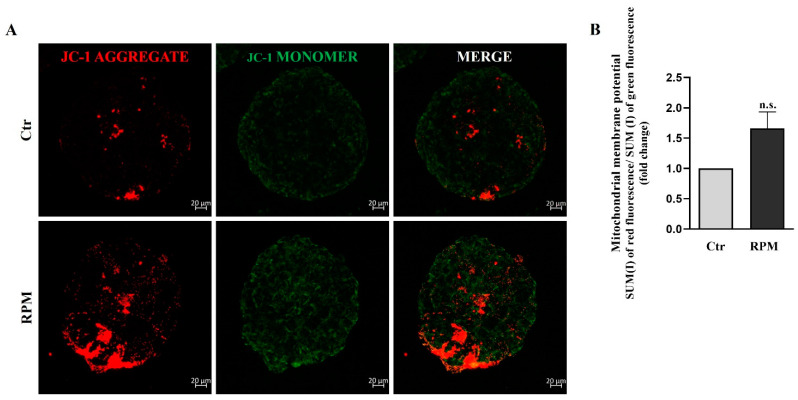
**Simulated microgravity induced a slight, but not significant, increase in mitochondrial membrane depolarization**. (**A**) Representative orthogonal projections of confocal analysis of TCam-2 spheroids, Ctr and RPM, after JC-1 staining (scale bar = 20 µm, n = 4 independent experiments). (**B**) The graph represents the ratio between the Sum(I) values of red fluorescence and green fluorescence (means ± S.E.M.; Student’s *t*-test, *p* = n.s., n = 4 independent experiments).

**Figure 6 cells-12-02106-f006:**
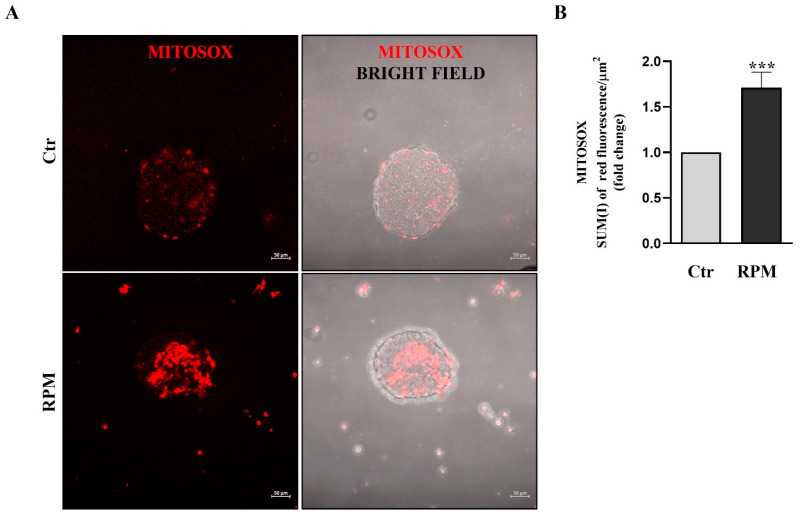
**Mitochondria superoxide production.** (**A**) Representative orthogonal projections of confocal analysis of TCam-2 spheroids in Ctr and RPM conditions after MitoSOX staining (scale bar = 50 µm). (**B**) Sum(I) of red fluorescence/area µm^2^ (means ± SE.M.; Student’s *t*-test, *** *p* < 0.001, n = 3 independent experiments).

**Figure 7 cells-12-02106-f007:**
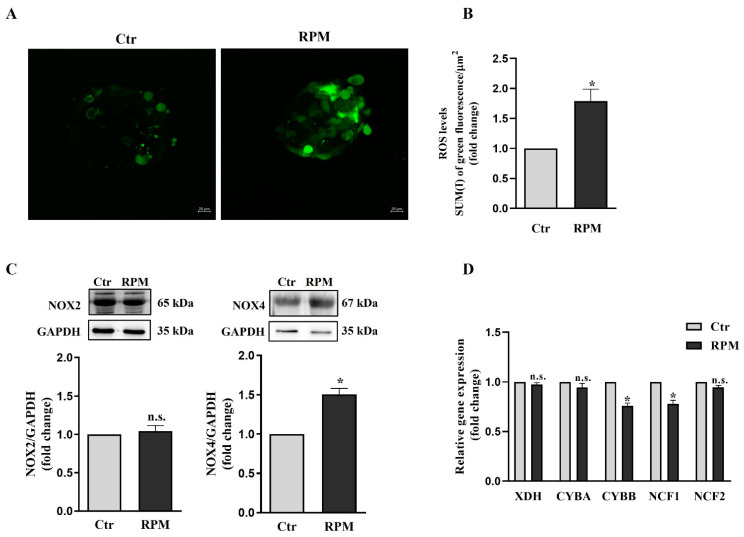
**ROS production and pro-oxidant enzyme expression profile.** (**A**) Representative confocal orthogonal projections of TCam-2 spheroids, cultured under Ctr or RPM conditions, after H2DCFDA staining (Scale bar = 20 µm). (**B**) Quantitative analysis of fluorescence after H2DCFDA staining (SUM(I) green fluorescence/AREA) (means ± S.E.M.; Student’s *t*-test, * *p* < 0.05, n = 3 independent experiments). (**C**) Representative immunoblots of NOX2 and NOX4 and the corresponding densitometric analyses. The densitometric values were calculated as the ratio between the optical density (OD) × mm^2^ of each band and the OD × mm^2^ of the corresponding GAPDH band, which was used as the loading control. The data are expressed as the ratio between the densitometric analysis of NOXs in cells exposed to s-microgravity (RPM) and the corresponding control cells (Ctr) (means ± S.E.M.; Student’s *t*-test, *p* = n.s., n = 6 independent experiments for NOX2; means ± S.E.M.; Student’s *t*-test, * *p* < 0.05, n = 3 independent experiments for NOX4). (**D**) qRT-PCR analysis of gene expression in TCam-2 spheroids exposed to s-microgravity (RPM) and unitary gravity (Ctr) conditions (means ± S.E.M.; One-way ANOVA test, * *p* < 0.05, n = 3 independent experiments).

**Figure 8 cells-12-02106-f008:**
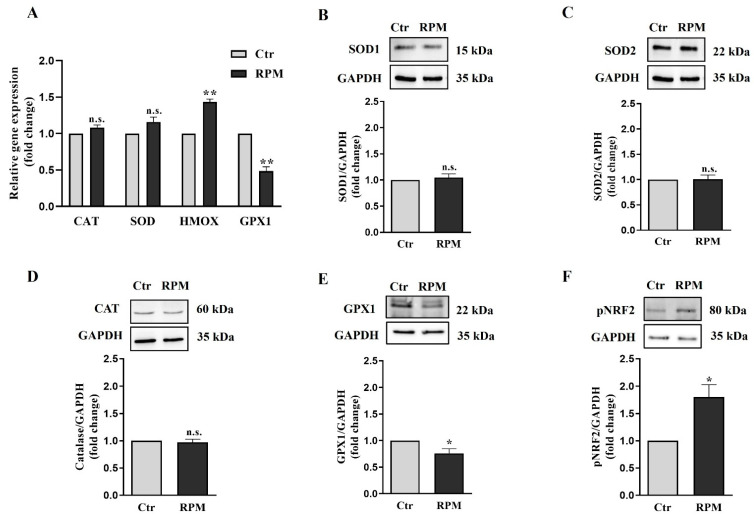
**Response of the antioxidant machine to s-microgravity**. (**A**) qRT-PCR analysis of gene expression in TCam-2 spheroids exposed to s-microgravity (RPM) and in control (Ctr) (means ± S.E.M.; one-way ANOVA test followed by the Bonferroni post hoc test, ** *p* < 0.01, *p* = n.s. n = 3 independent experiments). (**B**–**F**) Representative immunoblots of SOD1, SOD2, catalase, GPX1 and pNRF2, respectively, and the corresponding densitometric analyses. The densitometric values were calculated as the ratio between the optical density (OD) × mm^2^ of each band and the OD × mm^2^ of the corresponding GAPDH band, which was used as the loading control. The data are expressed as the ratio between results from samples exposed to s-microgravity (RPM) and those from the corresponding control (Ctr) (means ± S.E.M., Student’s *t*-test, * *p* < 0.05, *p* = n.s., n = 5 independent experiments).

**Figure 9 cells-12-02106-f009:**
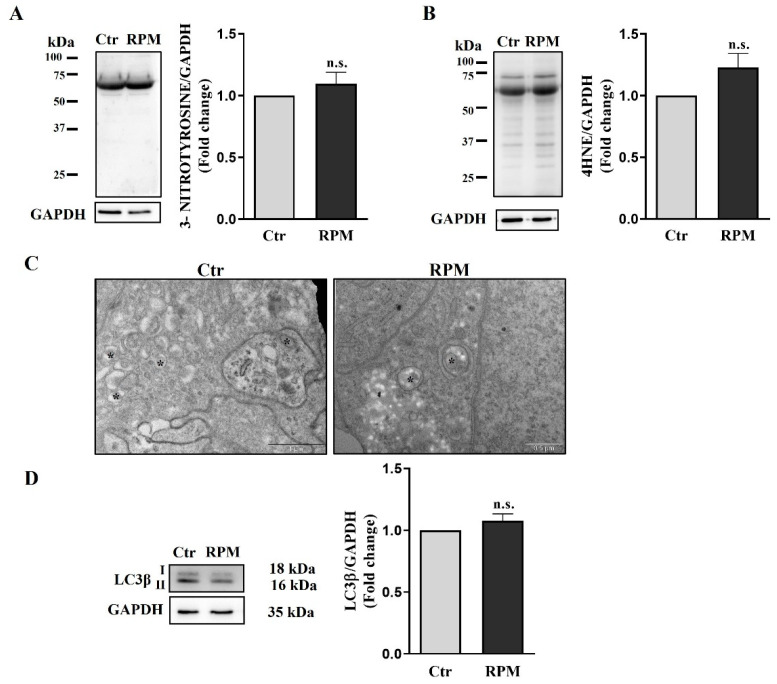
**Protein and lipid oxidative damage**. (**A**,**B**) Representative immunoblots of the expression levels of 3-nitrotyrosine and 4-hydroxynonenal (4HNE), respectively, and the corresponding densitometric analyses. The densitometric analyses were calculated as the ratio between the OD × mm^2^ of each band and the OD × mm^2^ of the corresponding GAPDH band, which was used as the loading control. The data are expressed as the ratio between results from samples exposed to s-microgravity (RPM) and those from the corresponding control (Ctr) (means ± S.E.M.; Student’s *t*-test, *p* = n.s., n = 7 independent experiments). (**C**) Representative images of ultrastructural analysis performed using TEM showing autophagic vesicles (asterisks) in Ctr (Scale bar = 1 µm) and RPM (scale bar = 0.5 µm) culture conditions. This analysis revealed that autophagy is active in both culture conditions. (**D**) Representative immunoblots of LC3β and the corresponding densitometric analyses. The densitometric analyses were calculated as the ratio between the optical density (OD) × mm^2^ of each band and the OD × mm^2^ of the corresponding GAPDH band, which was used as the loading control. The data are expressed as the ratio between results from samples exposed to s-microgravity (RPM) and those from the corresponding control (Ctr) (means ± S.E.M.; Student’s *t*-test, *p* = n.s., n = 3 independent experiments).

## Data Availability

Data supporting reported results can be obtained on request from the corresponding author. The whole Western blots are reported in a Appendix A.

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
