# Peer review of "Simulated Microgravity Exposure Induces Antioxidant Barrier Deregulation and Mitochondria Enlargement in TCam-2 Cell Spheroids"

_cells, 2023, doi:10.3390/cells12162106_

Round 1

Reviewer 1 Report

This Reviewer recommends publication with some revisions.

Please note the following comments:

-        Title: This Reviewer suggests to change the title by inserting "Simulated” ahead of microgravity, and specifying “mitochondria enlargement” instead of “alterations”.

-        Abstract: the abstract should contain a clearer explanation of why they in this work here the 3D model. This Reviewer thinks that "to gain more insights into the mechanisms of male germ cells response to altered gravity" is too generic. It would be enough to attach this paper to their previous one that used 2D monolayer. This information is given only at the end of the Introduction.

-        Introduction: the Introduction allows to follow the logical flow of the working hypothesis, provides a sufficient general background, and then presents in a descriptive way the main "actors" of oxidative stress, and of the antioxidant barrier that the Authors analyzed in their 3D model. It is this Reviewer’s opinion that, for sake of conciseness, "the results obtained…" can be removed from here, as they are pertinent to the Discussion section.

-        Materials and Methods: the methodology tackles the research questions appropriately. However, it is not explained why the Authors chose to expose the spheroids to 24 hours of simulated microgravity instead of 48 hours, as in their previous paper. In fact, 24 hours is quite a short timing, especially if consequences on the cell cycle are to be analyzed; even more so, if the study is to be put in the perspective of exploring the effects of long-term space missions, as stated by the Authors in the Introduction section.

The Authors should explain why they used 24 hours only, especially for a question of consistency when comparing the new results with previous data while changing the biological model.

-        Line 161-162, please correct the repetition of “mg protein mg protein”.

-        Line 260, paragraph “statistical analyses”: I suggest to specify also in the Figure legend where t-test and where one-way ANOVA were performed. Also, it is necessary to explicitly state how many times the experiments were repeated.

-        Results:

-        I suggest to state in all figure caption what statistical test was used.

-        Line 269, paragraph “T-cam2 spheroids viability”: fig.1A: state the magnification used in the caption, and/or the scale in the image more clearly.

-        Fig. 1B: the images are not in the same scale, the magnification should be evident or stated in the caption.

-        What does N=3 in the caption mean? Does 3 refer to experiments or fields?

Also, I suggest to indicate “not significant” (NS) when differences in the graphs are not.

-        Line 300, Caption of Figure 2:  What does “n=15” mean?

-        Please add Not significant (NS) where appropriate.

-        Line 316: Caption of Figure 3:  What does “n=6” mean?

-        Please add Not significant (NS) where appropriate.

-        Line 321, Paragraph “Mitochondria ultrastructure and functional features”: is it recommended to show statistical analysis to validate the statement of mitochondria enlargement. Since the information is shown in the title, it is important to corroborate the visual impression.

-        Line 366, caption of Figure 6: p<0,001 requires three stars of significance (as, correctly, in the graph).

-        Line 374: for p=0.012, adding two stars of significance is formally wrong. Please put one only.

-        Figure 7C: for a question of formal coherence, please fix the Y axes of NOS2/GAPDH graph.

-        Line 390 to 394: it is my impression that part of the figure legend ended in the text. Please correct.

-        Line 392: what does “n=6” mean? The same for n=3.

-        Line 397: the HMOX1 antioxidant gene appears upregulated in the graph, therefore please correct the typo “downregulated” that refers to this gene’s expression.

-        Legend of Figure 8: for the sake of clarity in the reader’s interest, I recommend to rewrite all the graphs just using the same scale.

-        Please note, in the first upper left graph, please add 1 to GPX (it should be GPX1).

-        Line 409: please change to ‘microgravity-induced’ (with final d).

-        Line 419: please clarify the meaning of “n=5”.

-        Finally, we would like to point out that while all paragraphs’ title are very concise, the title of paragraph 3.6 is more informative. We recommend to extend the latter style to all other previous paragraph titles because it is helpful to the reader.

-        Line 427: it is necessary to explain briefly why the LC3B marker was specifically chosen here. Also, since only one marker of authophagy is used, it is necessary at least to enlarge the body of references to support the choice. Therefore, I recommend to add references regarding the use of LC3B as authophagy marker in other cell models, at least one in simulated microgravity (Ae Jin Jeong et al., 2018, Scientific Reports 8 (1): 14646) and one in real Space flight (Barravecchia et al., 2021, Cell Mol Life Sci 79(1): 28).

-        Discussion: It is this Reviewer’s opinion that any conclusion on autophagy in the 3D model should be more cautious because the Authors analyzed LC3beta expression only. Moreover, it would be interesting to compare results from 2D and 3D samples, for a more informative analysis.

-        Also, the choice regarding timing should be discussed more. In fact, 24h of microgravity might not be enough to observe any significant change at the protein level.

-        Finally, it would be nice to find some comments about future work and future perspectives on the protection of human reproductive health in space environment.

Minor typos.

Author Response

Answers to the Reviewer 1

This Reviewer recommends publication with some revisions.

Answer: We wish to thank the Reviewer for the fruitful comments and amendments suggested to improve our manuscript. Herewith, we are submitting an updated and revised version, in which all the issues raised by the Reviewer have been addressed point by point, as follows.

Title: This Reviewer suggests to change the title by inserting "Simulated” ahead of microgravity, and specifying “mitochondria enlargement” instead of “alterations”.

Answer: According to the Reviewer’s suggestion we changed the title as follows: “Simulated-microgravity exposure induces antioxidant barrier deregulation and mitochondria enlargement in TCam-2 cell spheroids”.

-           Abstract: the abstract should contain a clearer explanation of why they in this work here the 3D model. This Reviewer thinks that "to gain more insights into the mechanisms of male germ cells response to altered gravity" is too generic. It would be enough to attach this paper to their previous one that used 2D monolayer. This information is given only at the end of the Introduction.

Answer: According to the Reviewer’s suggestion we modified the text of the abstract expanding on the reasons why we chose the 3D model (see track changes).

-           Introduction: the Introduction allows to follow the logical flow of the working hypothesis, provides a sufficient general background, and then presents in a descriptive way the main "actors" of oxidative stress, and of the antioxidant barrier that the Authors analyzed in their 3D model. It is this Reviewer’s opinion that, for sake of conciseness, "the results obtained…" can be removed from here, as they are pertinent to the Discussion section.

Answer: According to the Reviewer’s suggestion we removed the last sentence “the results obtained….” from the Introduction section.

-           Materials and Methods: the methodology tackles the research questions appropriately. However, it is not explained why the Authors chose to expose the spheroids to 24 hours of simulated microgravity instead of 48 hours, as in their previous paper. In fact, 24 hours is quite a short timing, especially if consequences on the cell cycle are to be analyzed; even more so, if the study is to be put in the perspective of exploring the effects of long-term space missions, as stated by the Authors in the Introduction section.

The Authors should explain why they used 24 hours only, especially for a question of consistency when comparing the new results with previous data while changing the biological model.

-Line 161-162, please correct the repetition of “mg protein mg protein”.

-Line 260, paragraph “statistical analyses”: I suggest to specify also in the Figure legend where t-test and where one-way ANOVA were performed. Also, it is necessary to explicitly state how many times the experiments were repeated.

Answer: Following the Reviewer’s suggestions, we clarified why we used 24h-exposure time (see track changes in the introduction, Materials and Methods and Discussion sections).

We have corrected the text accordingly to the Reviewer’s suggestions, adding in the figure legends the statistical analysis, and the number of experiments performed.

-           Results:

I suggest to state in all figure caption what statistical test was used.

-Line 269, paragraph “T-cam2 spheroids viability”: fig.1A: state the magnification used in the caption, and/or the scale in the image more clearly.

-Fig. 1B: the images are not in the same scale, the magnification should be evident or stated in the caption.

-What does N=3 in the caption mean? Does 3 refer to experiments or fields?

Also, I suggest to indicate “not significant” (NS) when differences in the graphs are not.

-Line 300, Caption of Figure 2: What does “n=15” mean?

-Please add Not significant (NS) where appropriate.

-Line 316: Caption of Figure 3: What does “n=6” mean?

-Please add Not significant (NS) where appropriate.

-Line 321, Paragraph “Mitochondria ultrastructure and functional features”: is it recommended to show statistical analysis to validate the statement of mitochondria enlargement. Since the information is shown in the title, it is important to corroborate the visual impression.

-Line 366, caption of Figure 6: p<0,001 requires three stars of significance (as, correctly, in the graph).

-Line 374: for p=0.012, adding two stars of significance is formally wrong. Please put one only.

-Figure 7C: for a question of formal coherence, please fix the Y axes of NOS2/GAPDH graph.

-Line 390 to 394: it is my impression that part of the figure legend ended in the text. Please correct.

-Line 392: what does “n=6” mean? The same for n=3.

-Line 397: the HMOX1 antioxidant gene appears upregulated in the graph, therefore please correct the typo “downregulated” that refers to this gene’s expression.

-Legend of Figure 8: for the sake of clarity in the reader’s interest, I recommend to rewrite all the graphs just using the same scale.

-Please note, in the first upper left graph, please add 1 to GPX (it should be GPX1).

-Line 409: please change to ‘microgravity-induced’ (with final d).

-Line 419: please clarify the meaning of “n=5”.

-Finally, we would like to point out that while all paragraphs’ title are very concise, the title of paragraph 3.6 is more informative. We recommend to extend the latter style to all other previous paragraph titles because it is helpful to the reader.

-Line 427: it is necessary to explain briefly why the LC3B marker was specifically chosen here. Also, since only one marker of autophagy is used, it is necessary at least to enlarge the body of references to support the choice. Therefore, I recommend to add references regarding the use of LC3B as authophagy marker in other cell models, at least one in simulated microgravity (Ae Jin Jeong et al., 2018, Scientific Reports 8 (1): 14646) and one in real Space flight (Barravecchia et al., 2021, Cell Mol Life Sci 79(1): 28).

Answer: As requested by Reviewer, we have modified point by point the text.

- In all figure captions what statistical test used was indicated

- In all figure captions “n” refers to the number of experiments performed. This information was added in the paragraph “2.8. Statistical Analyses” in the Material and methods.

- We apologize because the scale values are barely observable in figs 1A and 1B on the original images; for the sake of clarity, we added the bar value in all the figure captions.

- As suggested by the Reviewer all paragraphs’ titles were modified similarly to that of the paragraph 3.6.

- We agree with the reviewer that the enlargement of mitochondria needed to be deeper addressed and we added the measurement of mitochondria width in the Results section (see track changes in results section paragraph 3.4).

- We agree with the Reviewer that LC3B is only one of the key molecules involved in the complex mechanism of autophagy. In this study, we chose it because it was up-regulated in 2D TCam-2 cultures exposed to simulated microgravity for 24h (Morabito et al Sci Rep 2017). In addition, as stated by Reviewer, this marker was found up-regulated in other models exposed to simulated or real microgravity conditions. These considerations and references were added to the results in the paragraph 3.6.

- Finally, we added the remaining requested information in the Results and figure captions and graphs.

-Discussion: It is this Reviewer’s opinion that any conclusion on autophagy in the 3D model should be more cautious because the Authors analyzed LC3beta expression only. Moreover, it would be interesting to compare results from 2D and 3D samples, for a more informative analysis.

-Also, the choice regarding timing should be discussed more. In fact, 24h of microgravity might not be enough to observe any significant change at the protein level.

-Finally, it would be nice to find some comments about future work and future perspectives on the protection of human reproductive health in space environment.

Answer: We wish to thank the Reviewer for the suggestions. Let us highlight that even if we analysed LC3B only, our ultrastructural observations are in line with LC3B expression level confirming that autophagy is active in TCam-2 spheroids even at unitary gravity. Nevertheless, we agree with the Reviewer’s opinion, that the statements about autophagy of TCam2 spheroids need to be revised, since our results do not exclude that the active autophagic process could be used by TCam-2 spheroids to rapidly rescue the s-microgravity induced ROS-dependent cell damages, or that autophagy overactivation may occur at different times of s-microgravity exposure (see track changes in the discussion section).

- We clarified why we used 24h-exposure time (see track changes in the introduction, Materials and Methods and Discussion sections)

- We added a Conclusion paragraph mentioning future perspectives of this work.

Reviewer 2 Report

The submitted paper by Berardini M. et al. is a continuation of the work of this group of authors with the TCam-2 cell line when exposed to simulated microgravity. In this study, the authors assessed the state of mitochondria and participants in the ROS/antioxidant system balance. The results of the authors are of great interest, due to the choice of the object of study, not only for space biology and medicine, but also for fundamental studies of the behavior of malignant cells under various physical conditions.

Despite the high assessment of the work, I suggest that the authors pay attention to a number of points that I would recommend to finalize.

In the introduction, too much attention is paid to the elementary description of ROS and the antioxidant system, and not even in microgravity conditions. At the same time, the authors absolutely do not provide data on cellular respiration (OXPHOS) and ATP production in spermatozoa, in particular, during exposure under weightless conditions. Although such data reflect mitochondrial dysfunction. In addition, in line 321 of the results and in line 473 of the discussion, the authors show an increase in mitochondria. In the discussion, the authors should clearly explain what, in their opinion, the swelling of mitochondria may be associated with. In general, it is obvious that the goal of the authors was related to the balance of ROS and the antioxidative system, but I believe that it is necessary to compare their results with the literature data on the functioning of mitochondria in male mammalian germ cells under simulated microgravity.

A methodological note concerns loading control in western-blots. The authors postulate that they apply the same amount of protein per well and provide data for all replicas on the same membrane. However, gapdh in these replicas is very different, especially on membranes A, D, I, M. In addition, taking into account what is known from the literature about changes in cellular respiration, ATP production, and structural changes in mitochondria shown by the authors, the choice of gapdh as a reference is not optimal. Gapdh, glyceraldehyde phosphate dehydrogenase, is an enzyme of the Krebs cycle that takes place in mitochondria and therefore it can also change. Probably, the different levels of gapdh on the blots presented are associated with this. Usually, after the transfer, Ponceau staining is carried out, it is better to bring these data as a control of the same load.

As a minor point, the descriptions of the abbreviations when they are first used should be given: for example, in the abstract (line 42) there is the abbreviation SM, although s-microgravity is more common before and after.

Author Response

Answers to the Reviewer 2

The submitted paper by Berardini M. et al. is a continuation of the work of this group of authors with the TCam- 2 cell line when exposed to simulated microgravity. In this study, the authors assessed the state of mitochondria and participants in the ROS/antioxidant system balance. The results of the authors are of great interest, due to the choice of the object of study, not only for space biology and medicine, but also for fundamental studies of the behavior of malignant cells under various physical conditions.

Despite the high assessment of the work, I suggest that the authors pay attention to a number of points that I would recommend to finalize.

Answer to the general comment:

We wish to thank the Reviewer for the accurate revision and allow us to improve the quality of the manuscript. We provided a revised version in which we addressed all the points raised by the Reviewer.

In the introduction, too much attention is paid to the elementary description of ROS and the antioxidant system, and not even in microgravity conditions.

Answer: Trying to explain our point of view we wish to highlight that the focus on oxidative balance in this study is due to the strong published evidence that the increased ROS are one of main common intracellular target of changes in external mechanical forces, as it is microgravity. Indeed, the involvement of ROS balance in microgravity-induced effects was observed in different cell phenotypes from non- and mammalian origin (for example: Fanlei Ran et al Biophys Rep. 2016;2(5):100-105. doi: 10.1007/s41048-016-0029-0. Epub 2016 Nov 7; Thomas J. Goodwin et al Int J Mol Sci. 2018 Apr; 19(4): 959). In addition, this study represents the development of previous results demonstrating that simulated microgravity induced alteration of cell processes in TCam-2 monolayers via ROS-activated pathways, because the use of antioxidants prevents, almost completely, the microgravity induced effects (Morabito et al Sci Rep 2017). In the cells the sources of ROS are various, and their increase could be due not only to increased ROS production, but also to a failure in the physiological antioxidant/detoxifying systems of the cells. In this scenario, the mitochondria represent key organelles in the oxidative metabolic balance that also includes, or is included in, the cell metabolism. Thus, mitochondria represent not only the cellular energy powerhouse but also a system that dynamically buffers intracellular ions and molecules, as calcium and ROS (Edoardo Bertero and Christoph Maack Circ Res. 2018 May 11;122(10):1460-1478. doi: 10.1161/CIRCRESAHA.118.310082; Osellame, L. D., Blacker, T.S. & Duchen, M.R. Cellular and molecular mechanisms of mitochondrial function. Best. Pr. Res. Clin. Endocrinol. Metab. 26, 711–723, 2012).

At the same time, the authors absolutely do not provide data on cellular respiration (OXPHOS) and ATP production in spermatozoa, in particular, during exposure under weightless conditions. Although such data reflect mitochondrial dysfunction. In addition, in line 321 of the results and in line 473 of the discussion, the authors show an increase in mitochondria.

Answer: As far as spermatozoa is concerned, in the original text we mentioned some literature article and reviews on mammalian sperm (see references 20 and 21) and accepting the Reviewer’s suggestions we added some details on this model (see track changes of introduction section and reference 22). However, we wish to highlight that this study concerns the mitotically active male germ cells that are completely different, even at metabolic point of view, from the male gametes, and therefore a detailed description of microgravity exposure on sperm could be misleading.

In the discussion, the authors should clearly explain what, in their opinion, the swelling of mitochondria may be associated with. In general, it is obvious that the goal of the authors was related to the balance of ROS and the antioxidative system, but I believe that it is necessary to compare their results with the literature data on the functioning of mitochondria in male mammalian germ cells under simulated microgravity.

Answer: We agree with the Reviewer that the result of mitochondria morphology alteration deserves to be expanded a bit, even if we did not arrive at a definitive conclusion on the reason why this enlargement occurred. Therefore, accepting the suggestions of the reviewer, in the discussion we added a sentence with some speculations about the possible “meaning” of the presence of some swollen mitochondria (see track changes in the Discussion section).

A methodological note concerns loading control in western-blots. The authors postulate that they apply the same amount of protein per well and provide data for all replicas on the same membrane. However, gapdh in these replicas is very different, especially on membranes A, D, I, M. In addition, taking into account what is known from the literature about changes in cellular respiration, ATP production, and structural changes in mitochondria shown by the authors, the choice of gapdh as a reference is not optimal. Gapdh, glyceraldehyde phosphate dehydrogenase, is an enzyme of the Krebs cycle that takes place in mitochondria and therefore it can also change. Probably, the different levels of gapdh on the blots presented are associated with this. Usually, after the transfer, Ponceau staining is carried out, it is better to bring these data as a control of the same load.

Answer: The issue of loading markers is an important technical aspect always open when one begins a new study with new experimental conditions. Starting our studies on investigating the possible microgravity effects on cell models, we search for a convincing loading reference for Western blot analyses without excluding the Ponceau staining. Indeed, we used this last one to have a global check of the membrane loading, as it gives a non-specific staining, and rapidly wash the membranes because it can interfere with the chemiluminescence detection system. We excluded cytoskeletal proteins (as actin or tubulin, often used as loading control) because is well-known that microgravity significantly affected their expression in most of cellular models. So our choice was directed to GAPDH because, even in presence of modified metabolic behavior, the expression levels of this enzyme did not change in our conditions in different cell models (Guarnieri et al, Oxid Med Cell Longev. 2021, 2021:9951113. doi: 10.1155/2021/9951113; Catizone et al, Applied Sciences (Switzerland), 2020, 10(22), pp. 1–16, 8289; Morabito et al, Int J Mol Sci. 2020; 21(10):3638. doi: 10.3390/ijms21103638; Morabito et al, Int J Mol Sci. 2019; 20(8). doi: 10.3390/ijms20081892; Morabito et al, Sci Rep. 2017, 15;7(1):15648. doi: 10.1038/s41598-017-15935-z), this was also reported by other groups (for example: Ae Jin Jeong et al, et al., 2018, Scientific Reports 8 (1):14646; Thiel et al, Microgravity 2017; 3:22 ; doi:10.1038/s41526-017-0028-6; Xiaoyan Zhang et al, Front. Genet. 13:985025. Doi: 10.3389/fgene.2022.985025). Thus, we think that the changes observed in GAPDH levels, not reproducible, can be due more to a sample loading error than to a modulation of the enzyme by experimental conditions. Due to the importance of this technical aspect, we added in 2.4. Western Blot Analysis the statement regarding the choice of GADPH as loading control.

As a minor point, the descriptions of the abbreviations when they are first used should be given: for example, in the abstract (line 42) there is the abbreviation SM, although s-microgravity is more common before and after.

Answer: Thanks to the Reviewer, we changed “SM” (that was a typo) with s-microgravity. The whole text has been thoroughly edited and revised.

Reviewer 3 Report

Starting from the direction of "spaceflight", which is a hot topic in the contemporary society and even in the future, and connecting with the concern of reproductive health and genetic inheritance, the author's manuscript is a very interesting study. The results of the experiments are analyzed in depth on several occasions, but some parts of the manuscript need to be improved.

In the Introductory part, the narrative drags the focus slower, and it is suggested that the 1st paragraph can be brought to the theme of "male reproduction", and then introduce the specific scenarios and scientific background of microgravity, and then gradually expand to the impact of the TCam-2 cells and male germ cells.

In the Materials and Methods section, which details the experimental design process, I noticed that there didn't seem to be any data related to embryonic development, such as the development of the zygote resulting from the combination of exposed sperm cells and normal oocytes and the percentage of the zygote that grows to the next stage of development, which tends to be more intuitive to the intuitive effects of microgravity on the sperm cells as well as the results that one would want to focus on more than anything else.

In line 472, basically the experimental results shown are in vitro cultured and then derived, whether there is the possibility of other substances interfering with the growth of such a long period of time in between, rather than the influence of microgravity that the author wants to focus on, it is suggested that the author can make some more details on the description.

In line 487, this paragraph is described in a very detailed and logical way, starting with the ROS and then analyzing the enzymes, proteins and genes!

All in all, the author's research is very interesting, and it is very touching to see the attention paid to the very small number of people currently affected by microgravity, but who have made great contributions to the development of mankind, and will microgravity generate new ideas about the future development of microgravity? Overall this manuscript is very forward looking! I look forward to the author's next research!

Starting from the direction of "spaceflight", which is a hot topic in the contemporary society and even in the future, and connecting with the concern of reproductive health and genetic inheritance, the author's manuscript is a very interesting study. The results of the experiments are analyzed in depth on several occasions, but some parts of the manuscript need to be improved.

In the Introductory part, the narrative drags the focus slower, and it is suggested that the 1st paragraph can be brought to the theme of "male reproduction", and then introduce the specific scenarios and scientific background of microgravity, and then gradually expand to the impact of the TCam-2 cells and male germ cells.

In the Materials and Methods section, which details the experimental design process, I noticed that there didn't seem to be any data related to embryonic development, such as the development of the zygote resulting from the combination of exposed sperm cells and normal oocytes and the percentage of the zygote that grows to the next stage of development, which tends to be more intuitive to the intuitive effects of microgravity on the sperm cells as well as the results that one would want to focus on more than anything else.

In line 472, basically the experimental results shown are in vitro cultured and then derived, whether there is the possibility of other substances interfering with the growth of such a long period of time in between, rather than the influence of microgravity that the author wants to focus on, it is suggested that the author can make some more details on the description.

In line 487, this paragraph is described in a very detailed and logical way, starting with the ROS and then analyzing the enzymes, proteins and genes!

All in all, the author's research is very interesting, and it is very touching to see the attention paid to the very small number of people currently affected by microgravity, but who have made great contributions to the development of mankind, and will microgravity generate new ideas about the future development of microgravity? Overall this manuscript is very forward looking! I look forward to the author's next research!

Author Response

Answers to the Reviewer 3

Starting from the direction of "spaceflight", which is a hot topic in the contemporary society and even in the future, and connecting with the concern of reproductive health and genetic inheritance, the author's manuscript is a very interesting study. The results of the experiments are analyzed in depth on several occasions, but some parts of the manuscript need to be improved.

Answer: We wish to thank the Reviewer for the positive comment and the accurate revision of our manuscript.

In the Introductory part, the narrative drags the focus slower, and it is suggested that the 1st paragraph can be brought to the theme of "male reproduction", and then introduce the specific scenarios and scientific background of microgravity, and then gradually expand to the impact of the TCam-2 cells and male germ cells.

Answer: We wish to thank the Reviewer for the suggestion. Trying to explain the reason why we chose a different focus on the introduction section let us explain that, even if the influences of microgravity on male reproduction could be considered part of the goals of our research, this manuscript is mainly focused on the impact of s-microgravity on stem cells of male germ cells, instead of male gametes. However, partly accepting Reviewer’s suggestion we expanded the introduction section 1) adding a sentence on the importance of reproductive health of astronauts, 2) adding some details on the impact of microgravity on male gametes, 3) explaining the strategic reason of studying TCam-2 cells (see track changes in the Introduction section).

In the Materials and Methods section, which details the experimental design process, I noticed that there didn't seem to be any data related to embryonic development, such as the development of the zygote resulting from the combination of exposed sperm cells and normal oocytes and the percentage of the zygote that grows to the next stage of development, which tends to be more intuitive to the intuitive effects of microgravity on the sperm cells as well as the results that one would want to focus on more than anything else.

Answer: The goal of our research is the study of mitotically active germ cells that are the stem cells of the male gonad. Therefore, it seems to us that a description of embryonic development could be interesting when studying male gametes, but in our case could be misleading.

In line 472, basically the experimental results shown are in vitro cultured and then derived, whether there is the possibility of other substances interfering with the growth of such a long period of time in between, rather than the influence of microgravity that the author wants to focus on, it is suggested that the author can make some more details on the description.

Answer: We know that cell spheroids in the most cases develop using a specific “sphere medium” that contains different compounds with respect to media used for cell culture as monolayer. However, as described in the Materials and Methods section, the media and compounds used to culture TCam-2 spheroids are the same used in the previous paper to study TCam-2 cell monolayer. Therefore, the differences obtained cannot be ascribed to the differences in culture medium, but just to the different adhesive microenvironmental cues. We better specify this information in the Material and Methods section (see track changes.)

In line 487, this paragraph is described in a very detailed and logical way, starting with the ROS and then analyzing the enzymes, proteins and genes!

All in all, the author's research is very interesting, and it is very touching to see the attention paid to the very small number of people currently affected by microgravity, but who have made great contributions to the development of mankind, and will microgravity generate new ideas about the future development of microgravity? Overall this manuscript is very forward looking! I look forward to the author's next research!

Answer: We really thank the reviewer for the favourable opinion on our work, and we hope that we will not delude his/her expectation.

Round 2

Reviewer 2 Report

I believe that the authors have made significant corrections to the article, which greatly improved it. I think that the article can be accepted for publication.